# Brownfield Data and Database Management—The Key to Address Land Recycling

Lea Rebernik [1], Barbara Vojvodíková [2,*] and Barbara Lampič [1]

1 Department of Geography, Faculty of Arts, University of Ljubljana, Aškerčeva 2, 1000 Ljubljana, Slovenia
2 IURS—Institute for Sustainable Development of Settlements, 70800 Ostrava, Czech Republic
* Correspondence: barbara.vojvodikova@email.cz; Tel.: +420-725-117-244

**Abstract:** Brownfields sometimes represent a development problem but at the same time also hold development potential. With accurate and up-to-date information it is possible to assure the regeneration process is effective; therefore we investigated in detail the development of the process of brownfields management in two countries: Slovenia and the Czech Republic (the Moravian-Silesian Region). The article compares the process of development of databases and the data itself in both selected case studies, and evaluates and identifies the patterns of change in brownfields with a focus on regenerated sites. In the period 2017–2022 we have noticed a growing number of brownfields included in databases in both selected cases, despite the share of regenerated brownfield sites that have been excluded from the database. Both study cases show that ensuring continuity of work in the process of monitoring brownfields and knowledge transfer are critical for sustainable brownfield management and successful regeneration. Based on the comparisons, recommendations are summarised on how to make the database an effective tool that could be relevant to more sustainable brownfield development and land recycling.

**Keywords:** brownfield regeneration; change monitoring; database; sustainable development; Slovenia; the Moravian-Silesian region





## 1. Introduction

The European Union has set an ambitious goal for the future in the field of sustainable land use. It wants to achieve no net land take by 2050 [1], but Corine Land Cover data show that we are still far from this goal, as between 2000 and 2018 in the member states of the European Environment Agency (EEA-38) a total of 1000 km$^2$ of land has been built up each year [2]. Placing activities on previously developed land reduces the expansion pressure on agricultural and forest land, thus contributing to the achievement of sustainable spatial development goals. It follows rational (maximum positive effect of spatial activities) and efficient (appropriate planning, multipurpose use and the linking of sectors) spatial development and the goal of no net land take [1,3,4].

One of the most effective approaches to limiting the expansion of build-up land into new areas are planned efforts to make better use of the land. Efficient use of land (providing a mix of complementary land uses, supporting compact building design, and supporting regeneration) is considered the basis for sustainable development, and reuse of brownfields is promoted as one of the most important mechanisms to reduce the loss of fertile land and avoid land take [5–8].

The experience of various European countries in the field of brownfield reuse has shown many advantages and obstacles to its regeneration. Reuse of brownfields can be time-consuming, financially challenging, and often involves a high degree of uncertainty, especially for those responsible for change. In the long run, regeneration brings many economic, social, and environmental benefits, which are the mainstays of sustainable spatial development and have been pursued by all land-use policies and strategies in recent years.

Environmental benefits include, amongst other reducing expansion pressures, protecting public health and safety, protecting and conserving natural resources (groundwater, soil, etc.), eliminating environmental risks and restoring habitats. Social benefits include, for example, renovation and revitalization of town and village centres, enhancement of the quality of life, improvement of the appearance of the area, strengthening of local communities, and more. At the economic level, the value of land and real estate increases, new domestic and foreign investments are encouraged, the vitality of the city is improved, etc. [9–13].

Although brownfields can be a burden for different stakeholders, they also offer development potential. Foreign experience show that it is often difficult to identify, among the many characteristics of brownfields, those that have greater potential for regeneration. For example, a recent study [14] shows that the size of brownfields influences their new use—new functions. Recognizing the opportunities is essential for effective management. Brownfield regeneration is a complex task that must be considered as a priority, and solutions must be clearly structured and effective [6,15–18].

The article compares the methodological approach, development and management of brownfield databases from two European countries—Slovenia and the Czech Republic. Due to the difference in size between the two countries and the methodology of data collection (including existing databases) in the Czech Republic, we decided to make the comparison between Slovenia and a selected Czech region, the Moravian-Silesian region.

The decision was made to use the more widely known and common term "brownfield" throughout the article, although we are aware a modern definition of the term is needed, building on existing terminology. Human activities have changed drastically compared to the past, so new types of underused and disused sites are appearing in the space and we can no longer speak only of the "classic" brownfields (e.g., industry, mineral use and extraction, etc.).

The aim of the article is to (1) present (and evaluate) the process of the development of brownfield databases and the state of brownfields in Slovenia and the Moravian-Silesian region (Czech Republic) and (2) identify, analyse and evaluate the patterns of change on brownfields, focusing on the regenerated sites. By comparing data from 2017 and 2022, the article identifies transformation flows, new functions on regenerated brownfields and their relationship to the previous use. Understanding the relationships and changes is crucial to sustainable brownfield management and successful regeneration.

## 2. Materials and Methods

To truly understand the process of brownfield regeneration, a comparison was made between two geographic units. A direct comparison between Slovenia and the Czech Republic was not possible due to methodological obstacles, as (1) the size of the two countries (and thus the number of brownfields) is very different, and (2) the national database of the Czech Republic is only updated partially and not regularly, so complete and up-to-date data are only available at the level of some regions. To ensure the data quality and comparability of the data, the comparison was made between Slovenia and one of the 14 administrative regions of the Czech Republic—the Moravian-Silesian region (Figure 1). We are aware that these are two geographic units that have had different database and methodological development, thus this is the key to understanding good and bad practices. The reasons for the emergence of brownfields are very similar, both in the time period of their creation and the reasons for their creation. Both pilot areas have been trying to regenerate these sites for a long time. Data comparison in both pilot areas is relevant, as the analysed data focuses on the years in which both selected pilot areas already had an established database. Therefore, the data comparison is even more relevant.

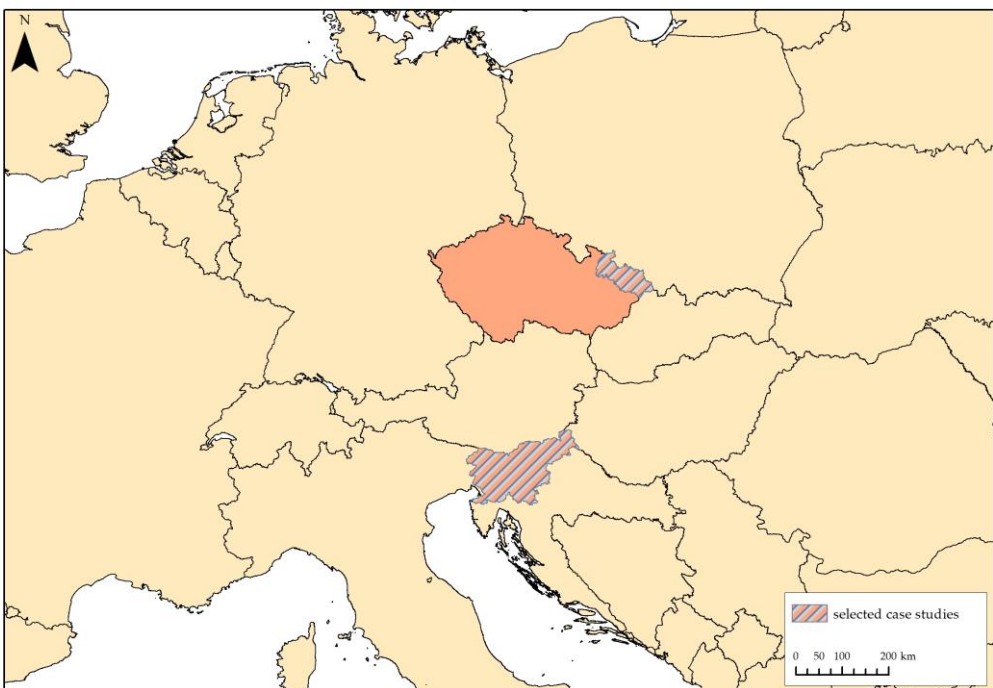

**Figure 1.** Map of both selected case studies—Slovenia and the Moravian-Silesian region (Czech Republic). Prepared by the authors [19].

Slovenia covers an area of 20,271 km$^2$ and is characterized by scattered and sparsely populated areas. The country is divided into 212 municipalities and has 2.1 million inhabitants (2020). The most densely populated municipality is Ljubljana (the capital), with an average of just over 1000 inhabitants per square kilometre [20,21]. More than half of Slovenia's land area is covered by forests (56% and 58%, respectively, including scrubland), while 34% of the national territory is used for agriculture [22].

The Moravian-Silesian region is located in the northeast of the Czech Republic. The region, with an area of 5427 km$^2$ (which is 6.9% of the total Czech territory), borders with Poland in the north and Slovakia in the east. With a population of 1.2 million (2021), it is the third most populous region in the Czech Republic. The region consists of 6 districts (Bruntál, Opava, Nový Jičín, Ostrava-město, Karviná and Frýdek Místek) and 300 municipalities. More than half of the territory of the region is occupied by agricultural land (more than 35% forest land). The whole region was one of the most important producers of coal and steel, but structural changes in the 1990s have affected the economy of the region and its landscape [23].

This article compares data for Slovenia and the Moravian-Silesian region from 2017 and 2022. In 2017, Slovenia created the first national database quantifying brownfields, so it is impossible to compare data from earlier periods. On the other hand, the data from 2022 (for both countries) represents the last update of both databases.

**The methodology was be divided into two parts**. In the first part, the two methodological approaches were compared in terms of brownfield definition, brownfield criteria, and brownfield typology. In addition, emphasis was put on technical aspects (structure and attributes) and the various functionalities of the databases. A study and presentation of the formal responsibility for the data and the updating process were evaluated and are presented. Finally, an up-to-date comparison of the usability of both brownfields databases was conducted, as well as an evaluation of interested stakeholders, the inclusion of the data in different documents, and their role in strategies, policies, and the spatial planning process.

The second part of the study focuses on the comparison of the current brownfield data (type, size, ownership, etc.) from 2017 and 2022, with special attention to the regenerated sites and their new function. (Figure 2).

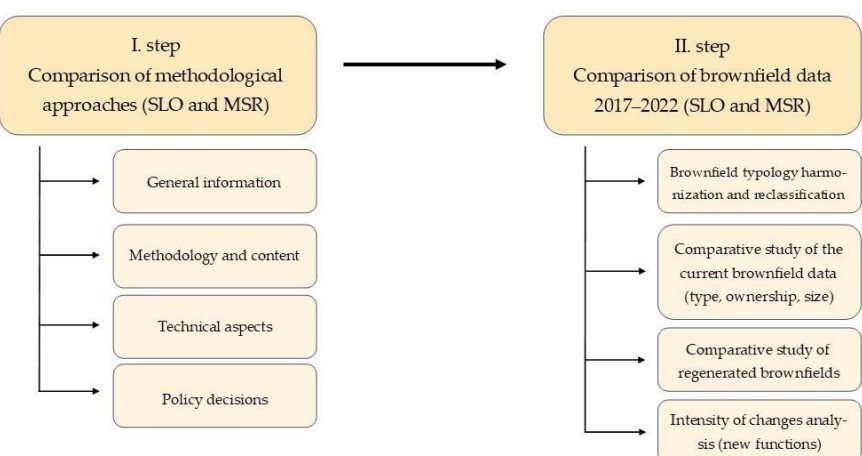

**Figure 2.** Methodological framework. Prepared by the authors.

The most important step of the research was to harmonize the typology of both databases. The Moravian-Silesian database distinguishes only between seven types of brownfields, while the Slovenian database distinguishes nine types (Table 1). After evaluating the data and comparing the typology, the decision was made to follow the Slovenian typology. The typology indicates the last activity before its suspension or the current prevailing activity at the site. In order to reclassify all Moravian-Silesian brownfields, the data on the previous activity of each site and the existing type (according to the Czech methodology) were determined and on this basis the new type was assigned.

**Table 1.** Slovenian brownfield typology as the basis for the comparison [5].

| Nr. | Brownfield Types |
|:---:|:---:|
| 1. | agricultural activities |
| 2. | commercial and service activities |
| 3. | tourism, hospitality and sport |
| 4. | industrial, craft and storage activities |
| 5. | defence, protection and rescue services |
| 6. | mineral extraction and use |
| 7. | infrastructure |
| 8. | transitional use |
| 9. | housing |

In order to further analyse the current state of brownfields and their regeneration in Slovenia and the Moravian-Silesian region, comparative studies were conducted in the second step. For each brownfield site, we collected and compared data on its type, ownership, size and new function. These data form the basis for identifying the flows of changes taking place on the sites.

The primarily focus of the analysis was on regenerated brownfields and their new use or transformation, particularly from the type of brownfield to the type of new function at the site. It should be emphasized that in conducting the analysis, two types of sites were considered—(1) sites that have already been fully regenerated (are back in use) and (2) sites where intensive construction is underway and are expected to be regenerated within a 2-year period (meaning that the investor and the plans for regeneration of the site are known).

Based on the description of new activities at each site, a broader typology of new functions was first created. Due to the wide variety of new functions occurring on regenerated

sites, it was decided to group similar categories together and simplify the classification system. In developing the typology of new functions on brownfields, the goal was to have fewer types in terms of number that are structurally comparable to the typology of brownfield. The typology includes nine types of new function (Table 2) and differs slightly from the brownfield typology.

**Table 2.** Types of new functions.

| New Function | Description/Examples |
|---|---|
| Agricultural | Buildings with functional land intended for living, agricultural food production and accompanying activities related to agriculture. Only agricultural land (forest, grassland, etc.). Other agricultural or forestry activities, hunting (e.g., fish farms, greenhouses, etc.) |
| Green areas | e.g., parks, communal green areas, children's playgrounds, etc. |
| Commercial and services | Areas with buildings and functional land for business, service and trade activities. e.g., technological parks, business zones, business facilities, shops and supermarkets. |
| Public services | Areas with facilities and functional land for educational, health, cultural, religious or other public activities. e.g., schools, houses of culture, health centres, residential care homes, etc. |
| Defence, protection and rescue | e.g., barracks, warehouses for the army, installations and training ranges, missile systems (missile bases) for the defence of airspace, shooting ranges, patrols, etc. |
| Housing | e.g., apartment buildings, single family dwellings. |
| Industrial, craft and storage | e.g., storage premises, industrial and craft zones. |
| Infrastructure | Areas for production, transmission and distribution of energy products—e.g., biogas plants, photovoltaic power plants, etc. Areas of surfaces for transport—e.g., parking lots, railway and bus stations, logistics and transport terminals, etc. |
| Mineral extraction and use | e.g., quarries, sand pits, etc. |
| Tourism, hospitality and sport | Areas with facilities and functional land for tourist activities, sports and/or recreational activities. e.g., hotels, motels, inns, restaurants, guesthouses, swimming pools, football stadiums, tennis, football, basketball and other sports fields, sports hall, etc. |

Based on [5]. Prepared by the authors.

Joined categories provided greater insight into the changes in activities that occurred at the regenerated sites. The final step of the analysis was devoted to the intensity of the changes. The Cytoscape 3.9.1. (Institute of Systems Biology, Seattle, USA) programme was used to graphically display the flows of change and, more importantly, to display the intensity of change from each individual type of brownfield to each individual new function. This is key to analysing spatial changes and helps to assess whether or not brownfield regeneration was sustainable and whether or not the new use of the site is appropriate.

*General Characteristics of Brownfields in Selected Case Studies*

In 2022, 1154 brownfields with a total area of 3723.5 ha were recorded in Slovenia. Compared to 2017, we can note a slight increase in the number and total area (1081; 3422.7 ha). The inventory showed that, by number, brownfields of industrial, craft and storage activities predominate (219), followed by brownfields of mineral extraction and use (180) and brownfields of commercial and service activities (178). The total area of brownfields for industrial, craft and storage activities is 1129.4 ha, followed by brownfields of mineral extraction and use (787.2 ha) and brownfields of infrastructure (501.9 ha).

Between 2017 and 2022, the number of brownfields in the Moravian-Silesian region increased (from 537 to 739), but on the other hand, their total area slightly decreased (from 1563.5 ha to 11,513.8 ha). A closer look at the structure of brownfields reveals a clear difference with Slovenia. In the Moravian-Silesian region, brownfields of agricultural activities predominate (156), followed by brownfields of defence, protection and rescue services (138) and brownfields of commercial and service activities (126). In terms of total area, brownfields of defence, protection, and rescue services (524.9 ha) and brownfields of industrial, craft and storage activities (379.9 ha) together occupy more than half of the total area, followed by brownfields of agricultural activities (202.4 ha). This indicates that the structure of brownfields in the Moravian-Silesian region is a clear consequence of the historical development of the region as one of the most important coal and steel producers in the Czech Republic and its current development, as more than half of the territory of the region is occupied by agricultural land (Table 3).

**Table 3.** Number and total area of brownfields in Slovenia and the Moravian-Silesian region in 2017 and 2022 (by brownfield type).

| Brownfield Type | SLOVENIA | | | | MORAVIAN-SILESIAN REGION | | | |
|---|---|---|---|---|---|---|---|---|
| | 2017 | | 2022 | | 2017 | | 2022 | |
| | nr. | Total Area (ha) | nr. | Total Area (ha) | nr. | Total Area (ha) | nr. | Total Area (ha) |
| Brownfields of agricultural activities | 75 | 202.3 | 84 | 245.6 | 117 | 180.7 | 156 | 202.4 |
| Brownfields of commercial and service activities | 171 | 324.3 | 178 | 360.2 | 65 | 37.9 | 126 | 50.2 |
| Brownfields of tourist, hospitality, sports and recreation activities | 60 | 102.2 | 76 | 129.6 | 15 | 9.2 | 44 | 38.3 |
| Brownfields of industrial, craft and storage activities | 228 | 1196.9 | 219 | 1129.4 | 105 | 505.3 | 106 | 379.9 |
| Brownfields of defence, protection and rescue services | 34 | 152.1 | 35 | 164.5 | 130 | 639.8 | 138 | 524.9 |
| Brownfields of mineral extraction and use | 171 | 649.9 | 180 | 787.2 | 7 | 21.2 | 11 | 111.2 |
| Brownfields of infrastructure | 128 | 418.4 | 159 | 501.9 | 21 | 145.5 | 20 | 174.0 |
| Brownfields of transitional use | 116 | 267.8 | 142 | 307.2 | 3 | 0.7 | 4 | 3.2 |
| Brownfields for housing | 98 | 108.8 | 81 | 97.9 | 74 | 23.2 | 134 | 29.7 |
| **total** | **1081** | **3422.7** | **1154** | **3723.5** | **537** | **1563.5** | **739** | **1513.8** |

Based on [24–27]. Prepared by the authors.

From a regeneration perspective, it is important to understand the structure of brownfield sites (by type). Most of these sites have a major impact on the environment, and on past and present pollution from their activities, and therefore require a different approach to regeneration.

## 3. Results and Discussion

Brownfields are defined differently in different countries [28–31], but at the global level there is no organization or initiative working on collecting methodologically comparable data on types of brownfields, regeneration processes, data availability, etc. [32]. Therefore, the definition of brownfields depends on the purpose and objectives of an individual study, and in most cases the challenges of regeneration are left to regulations at the regional and/or national level [5].

The two selected case studies have gone through a different development process of the brownfields database and have been in operation for different lengths of time. Despite the fact that both have gone through a different development path and that brownfield identification in Slovenia is a more recent topic, we can identify some commonalities (e.g., the importance of research projects for the development of the methodology, the irreplaceable role of fieldwork, data collection, etc.).

Table 4 provides insight into eight points of comparison that can be divided into four groups—general information, methodology and content, technical aspects, and policy decisions—providing important insight into comparing data and ensuring proper data interpretation. The comparison highlights the strengths and weaknesses of the two methodologies, the databases, and other specific elements of the comparison. The results serve as a tool for future improvements to existing databases, for more sustainable and long-term management of brownfields, or as an advisory tool for those whose methodology and database are currently under development.

**Table 4.** Comparison of Slovenian and the Moravian-Silesian brownfield data and database in eight points of comparison.

| Points of Comparison | Slovenia | Czech Republic/Moravian-Silesian Region (MSR) |
|---|---|---|
| Database establishment, financing and maintenance | • 2010–2012 The first systematic inventory of 4 selected brownfield types in Slovenia was carried out as a part of a wider project on *Sustainable remediation of environmental burden in Slovenia*, supported by the Slovenian Research Agency and the Ministry of Economic Development and Technology.<br>• In 2015 a regional pilot project was carried out. The students from the Department of Geography (Faculty of Arts, University of Ljubljana) built on the existing typology of brownfields, identified, and spatially defined brownfield in Gorenjska statistical region. This represents a basis for the future national database.<br>• 2015–2017 The Slovenian national database was created as a result of a national target research programme project, co-financed by the Slovenian Research Agency and Ministry of Economic Development and Technology. Methodology was developed by the University of Ljubljana (Department of Geography, Faculty of Arts and Faculty of Civil and Geodetic Engineering).<br>• 2019–2020 the first official update was completed, financed by Ministry of Economic Development and Technology (updated by the Department of Geography, FA UL).<br>• 2020–2022 The Department of Geography (FA UL) carried out continuous updating and field work on its own.<br>• In 2021 the Department of Geography (FA UL) created the first database of potentially contaminated sites based on the brownfields database (financed by the Ministry of Environment and Spatial Planning).<br>• 2022–2023 The second official update was carried out by the Department of Geography (FA UL) (financed by the Ministry of Environment and Spatial Planning). | • The first database in the Moravian-Silesian region was the Ostrava database (2000). The database is a result of a research project supported by the Ministry of Industry.<br>• 2005–2006 The National Brownfields Database was prepared at the state level—the project *Brownfields 3000*. It was supported by the Ministry of Industry and contained only sites potentially suitable for industrial use, bigger than 0.5 ha (only one site from the Ostrava database was included).<br>• In 2013 ARR (Regional Development Agency) took the national data and created a regional database, but it did not include the data from the Ostrava database.<br>• In 2014 a database update was carried out by the students from the VŠB Technical University of Ostrava (VŠB TUO) (financed by ARR). The project introduced some modifications to the structure of the information about the sites.<br>• 2015–2016 Mapping continued through the cooperation of ARR and the VŠB TUO (financed by ARR).<br>• In 2016 ARR took over most of the responsibility for maintaining the database.<br>• In 2017 the whole database was updated (financed by ARR). For the first time the information was linked with the Contaminated site registration system [33].<br>• 2017–2019 within the framework of the *LUMAT project* (Interreg Central Europe), the database was updated and more intensively linked with SEKM (ARR was renamed to MSID).<br>• 2020–2022 MSID carried out continuous updating of the database in cooperation with municipalities. In cooperation with the GIS department at MSR, brownfields were included in the MSR map portal. |

**Table 4.** *Cont.*

| Points of Comparison | Slovenia | Czech Republic/Moravian-Silesian Region (MSR) |
|---|---|---|
| Brownfield definition | • Spatial Planning Act definition: "An area which has been, due to inappropriate or abandoned use, reduced in economic, social, environmental, visual or heritage value and needs redevelopment; the area may exhibit different types and degrees of degradation according to physical, functional, environmental, social and heritage criteria" [34].<br>• Narrowed formal definition from the Spatial Planning Act for the first database creation: "Not fully utilized or disused areas with a visible impact of its former uses and of lower utility value, that needs to be regenerated" [5,35]. | • "A property (area, site, land, building) that is unused, neglected and may be contaminated. It arises as a relic of industrial, agricultural, residential, military, or other activity. Brownfield cannot be used appropriately and effectively without a process of regeneration" [36].<br>• There is no formal, legislative definition. |
| Brownfield criteria | • In 2012 we differed only between 4 types of brownfields. Minimum size was 1 ha.<br>• The key criteria in the current database is the absence of activity (abandonment) or disused sites where the activity is reduced (completely abandoned, predominant abandoned, partially abandoned—but at least 10%).<br>• The minimum size is 0.5 ha, while in urban settlements it is 0.2 ha.<br>• The border of the area runs along the plot boundaries (land cadastre). | • 2005–2006 The key criteria were the potential suitability for industrial use and a size over 0.5 ha.<br>• Since 2014, the MSR databases has no minimum size rules.<br>• In addition to meeting the definition, the critical factor is that the site presents a problem, and its regeneration would bring positive benefits to the municipalities and their residents.<br>• The border of the area runs along the plot boundaries (land cadastre). |
| Brownfield types | • The typology is developed based on the former activity or on the latest activity before its suspension. 9 types are defined, with 8 of them also being further classified into subtypes (23 subtypes).<br>• Brownfield types: agricultural activities; commercial and service activities; tourism, hospitality, sports and recreation activities; industrial, craft and storage activities; defence, protection and rescue services; mineral extraction and use; infrastructure; transitional use; housing. | • Brownfields are divided into 7 types (industry, agriculture, landfill, military, traffic, civil facilities and others). |
| Database structure and attributes | • The Slovenian database includes more than 40 attributes (numerical and descriptive):<br>  1. Identification of the brownfield (identification number, name, type/subtype, size, ownership, contact person etc.);<br>  2. Location (municipality, region);'<br>  3. Abandonment (abandonment rate, year of construction and year of abandonment);<br>  4. Physical conditions (status and description of buildings and surrounding area maintenance);<br>  5. Environmental degradation (potential contamination etc.);<br>  6. Future development (development plans, obstacles, and timeframe of the envisaged reactivation activities);<br>  7. Detected changes (type of changes, year of detected changes and detailed description of changes);<br>  8. Other (attached photos). | • The MSR database includes 30 attributes:<br>  1. Identification of the brownfield (identification number, name, size, ownership, contact person etc.);<br>  2. Location (coordinates, cadastral parcel number, municipality etc.);<br>  3. Environmental degradation (priority number, expected contamination etc.);<br>  4. Data on the locality (number of buildings, connection to utilities etc.)<br>  5. Connection to road infrastructure (highways);<br>  6. Analytical data (abandonment rate, priority category etc.);<br>  7. Future development. |
| Data collection and updating process | • In 2017 the data was obtained with target interviews (municipality representatives—including all 212 Slovenian municipalities) and direct field visits (of all brownfield sites in the country). The collected data were entered into a specially designed application (including photos of each site), sites (polygons) were georeferenced.<br>• The 2019–2020 and 2022–2023 updates conducted using a participatory approach including information from all municipality representatives (a tool, web application in ArcGIS Online for data collection designed), controlled and centrally reviewed by the Department of Geography (FA UL). Where necessary, additional field visits and final data verification conducted on specific sites [37].<br>• Permanent continuity of updating is assured with publicly available information sources (other public databases, media etc.) (by the Department of Geography, FA UL).<br>• In the future it is expected that the updating cycle will be carried out in 3-year cycles. | • Until 2017, the data were collected mainly with field surveys (basic word documents with brownfields sheets and photo documentation). Later the data were transcribed (often done by students working at ARR) which resulted in errors (especially location). Eventually cadastral land numbers were added to the description of the site, to avoid mistakes.<br>• From 2017 on, updating is the responsibility of MSID, with the occasional help of students of VŠB TUO. Updating and supplementing is done in cooperation with municipalities and with the help of publicly available information sources. Field visits to specific sites are made to verify the accuracy of the information and to prepare photo documentation.<br>• The data is collected in an Excel document. Based on a corrected and revised files, a map was prepared at the MSR map portal. The main document and the database are being updated continuously and systematically.<br>• The web map is updated once a year on the basis of MSID information. |

Table 4. *Cont.*

| Points of Comparison | Slovenia | Czech Republic/Moravian-Silesian Region (MSR) |
|---|---|---|
| Data availability | • The first database has been publicly available since the first national project (2017–2020) [38].<br>• Since 2021 the database is included into *Spatial information system of municipalities* [39].<br>• The data are available on several portals, where different displays are used, and only limited attributes are available (15). The portals always display the latest official data set [40–43].<br>• 2022–2023 A project has started which will ensure that the database is included in e-Space (*Slovenian e-Prostor*) thus becoming an administrative spatial record [44]. | • There are several portals available—e.g., Invest more (available since 2016, but not updated since 2018, it only contains sites with owner's permission), the LUMAT project portal (contains several localities and examples of regeneration). All are outdated and not actively maintained but are still accessible online.<br>• The data is currently available on MSR Map Portal: https://geoportal.msk.cz/Public/Apps/brownfield/index.html (19 September 2022). |
| Inclusion in official documents | • The data (and the approach) are included in various strategic and policy documents [45,46].<br>• The *Spatial Planning Act* [34] defines brownfields and sets their regeneration as a priority. Brownfields are identified as an important resource in achieving rational land use.<br>• In the *Investment Promotion Act* [47], one of the criteria for granting incentives is that the site of the investment is a brownfield with an appropriate planned land use or in an existing craft business zone.<br>• Included in *Draft of the Spatial Development Strategy of Slovenia 2050* [3].<br>• An environmental indicator *Functionally degraded areas* [48] is included in the Environment indicator system of the Slovenian Environmental Agency. | • The *Building Act* [49] states: "to create conditions in the territory for eliminating the consequences of sudden economic changes, especially by examining and possibly defining buildable areas or transformation areas". A transformation area is an area intended to create a completely new character of the territory or to restore a degraded area for the purpose of its reuse, defined in the built-up area by the master plan.<br>• In the *National Strategy for Brownfields Regeneration 2019–2024* [35]. One of the tasks is the updating of the national database of brownfields and its connection with relevant data on the territory.<br>• The *Regional Development Strategy of the Czech Republic 2021+* includes revitalising brownfields as a part of strategic objective 3.<br>• Included in the *Action plan for the regeneration of brownfields* in the Moravian-Silesian region. |

Prepared by the authors.

### 3.1. Results of the Comparison of the Slovenian and the Moravian-Silesian Brownfield Data and Database Development and Its Present Function

Although we compare the national level (Slovenia) with the regional level (the Moravian-Silesian region), the identification of brownfields in each Czech region remains independent, which allows the comparison without compromising objectivity. Nonetheless, understanding the national level in the Czech Republic is important and is therefore included in the analysis where relevant.

The Czech Republic has been one of the leading countries in Europe in dealing with brownfields for twenty years. This is reflected in numerous scientific articles [50–54] and professional publications, as well as in projects that have paved the way for brownfields research. The well-known national project *Brownfields 3000* (2005–2006), supported by the Ministry of Industry, facilitated the process of data collection at the national level. The same applies also for Slovenia, where a project (co-funded by the Slovenian Research Agency) was crucial for the development of the methodology and the first database also in Slovenia (2017). Even though the Czech Republic has a much longer history in brownfields research and targeted activities related to brownfields, the late start in monitoring brownfields was an advantage for Slovenia. This is mainly due to the wider use of geographic information systems (and various tools/apps) and better networking of available spatial data.

Funding and maintenance of the databases is different in the two selected cases. With the change of responsibility in Slovenia to the Ministry of Environment and Spatial Planning, the financing of the database and data maintenance is formally guaranteed (the database is included in "*e-Space*" and thus becoming an administrative spatial record). In the Moravian-Silesian region the current funding is less stable and not well defined. MSID conducts ongoing updates with its own and regional funding sources.

Table 4 clearly shows the differences in definition, especially within legislation. In Slovenia, the general definition of a brownfield site (*Slovenian "razvrednoteno območje"*) is included in the law [34]. This makes it an official term that can be worked with and referred to. However, for the purpose of creating a brownfields database, the legal definition was somewhat narrowed. The additional criteria and the brownfield typology ensured the

visibility and enforcement of the brownfield database, which is why the term was changed several times (in legal documents, etc.).

In the Czech Republic, the definition is not regulated by law. Each region or each institution providing subsidies slightly modifies the basic definition based on the CABERNET network. The reasons for this are complex. One of the first obstacles was the prohibition of using English terms in Czech legislation. This led to the emergence of many formulations that tried to copy the Czech term in Czech words. Nowadays, however, brownfield is recognized as a Czech word and is used as a Czech word when writing sentences. The second reason is the historical association of the word brownfield with heavily polluted land. The definition given in the table is a consensus of all ministries involved in preparing the National Strategy for Brownfield Regeneration. In the Moravian-Silesian region, identification was carried out according to a very similar definition.

In Slovenia, the main criterion for including a site in the current database is the absence of activity (abandonment). The sites where activity is reduced (fully or partially abandoned) are included. The second criterion is the minimum size (0.5 ha or 0.2 ha in urban settlements). In the Czech Republic, the minimum size has been gradually abandoned. The key to the inclusion of small sites in the database is whether they pose a problem for their surroundings. Such an approach is controversial, since it is based on a subjective decision.

A comprehensive database structure and attributes in both study areas include numeric and descriptive attributes, and some data are also taken from other public databases. The Slovenian database includes more than 40 attributes, while the Moravian-Silesian database includes 30 attributes. However, in our experience, the quality of the data obtained is more important than the number of attributes. Especially in the case of quantitative data, descriptions and field findings (and photos) are irreplaceable. In a period of seven years of data collection and updating, the brownfields database in Slovenia has provided very rich, including comprehensive information about each brownfield site, enabling us to understand the processes taking place on the site.

The method of data collection and updating is very similar in both study areas. It is based on research, obtaining data from municipalities and field surveys. We must emphasize the importance of field work and linking it to actual conditions and the process underway—verifying the situation in the field. The main difference is in the subsequent data processing technology. Since Slovenia started creating the database much later, it did not have to go through the lengthy process of changing from paper format to an Excel spreadsheet. This also eliminates the problems described in the Moravian-Silesian database (Table 4), which were mainly caused by the step-by-step manual transcription.

Data availability is ensured in both countries, but access to all attributes, photos, and spatial data of brownfields for further analysis is limited. There is a clear and significant difference in access to the official databases in the two case studies. In Slovenia it is already part of a systematic process supported by national funds (but by different ministries). At the moment, it is based on a common definition and technically supported by GIS, as the Department of Geography (FA UL) has been taking care of collecting and updating data from the very beginning. In this way, continuity of work and continuous improvement of the monitoring system is ensured. In the Czech Republic, the database has developed gradually, without unified support, definition of a hierarchy, unified methodology and without IT support. However, it should be emphasized that the Czech Republic and the Moravian-Silesian region started developing the database 20 years earlier. The current inconsistency is due, among other things, to the inconsistent funding system.

The problem in the Czech Republic was the existing conflict of responsibilities between the Ministry of Environment, the Ministry of Industry and Trade, the Ministry of Regional Development and the Ministry of Agriculture. In Slovenia, the issue of shared dilemma regarding the responsible ministry was resolved in 2021, when the brownfields database was taken over by the Ministry of Environment and Spatial Planning. The database has been integrated into the geographic information system of municipalities [39] since 2021, so that the basic 15 attributes and spatial layers are available to various users and stakeholders.

The data and photos of the brownfields are also available online in an ArcGIS Online application (dashboard). Some other portals have also included the database, and starting in 2023, the database will have administrative geospatial dataset status.

Finally, we also explored the forms of the theme and the inclusion of the data in different strategies and policies. Slovenia was able to introduce the topic in various documents in a very short period of time, starting in 2017. The key to success was an overarching strategic document, the National Development Strategy 2030, which facilitated the implementation of the content in regional policies. Although the topic of brownfields is included in the Spatial Planning Act and the Investment Promotion Act, a proper action plan for the regeneration of brownfields is missing. On the other hand, there is a long tradition of dealing with brownfields in the Czech Republic, which is reflected in various documents. In the Moravian-Silesian region, the action plan for the regeneration of brownfields is being implemented.

The Slovenian database helps to improve the results in the field of brownfields regeneration, as it is continuously updated, contains more (also more accurate) data, and because its design allows better connectivity with other databases (and quicker analysis). Since the database has only been in existence since 2017, which is a short period, it is difficult to monitor its effectiveness, thus we can expect this to be possible in the years to come. Despite this, the database is only one of the tools in brownfield regeneration, which needs to be considered when monitoring the regeneration effectiveness. Better and more accurate data gathering and management means time saving and quicker availability. With that said, we also cannot state that the Czech Republic has a reduced brownfield regeneration or that its regeneration is ineffective, just because the database is not well designed. What Slovenia lacks, on the other hand, is the main advantage of the Czech Republic—an established system, a longer history of monitoring changes and an existing action plan for brownfields regeneration, which makes them at least one step ahead of Slovenia in the comprehensive management of brownfields.

### 3.2. Results of the Comparison of Slovenian and the Moravian-Silesian Brownfields Structure

A more detailed comparison of the structure of brownfields by type in Slovenia and the Moravian-Silesian region (Figure 3) shows that there are large differences between certain types. This is due to different historical development as well as different methodology of data collection and brownfields identification in both case studies and should be considered when analysing the data. For example, certain (sub)types of brownfields, such as transitional use or environmental infrastructure (which include illegal landfills), are not identified in the Moravian-Silesian region. Also, former quarries and sand mines are not recorded in the database as brownfields of mineral extraction and use, so the final figures for all the above mentioned brownfield types are much lower than in the Slovenian example.

There is a significant difference in the structure of brownfields according to the total area (Figure 4). In the Moravian-Silesian region the brownfields of defence, protection and rescue clearly dominate, while in Slovenia it is the brownfields of industrial, craft and storage activities. Some differences can also be observed, both in terms of number and total area. For example, in Slovenia there are fewer brownfields for housing (81) than in the Moravian-Silesian region (134), but on the other hand, the total area of Slovenian sites (97.9 ha) is three times larger than that of the Moravian-Silesian region (29.7 ha). There are two main reasons for these results. First, the difference is due to the minimum size criteria. In Slovenia there is a minimum size (0.2 or 0.5 ha), while in the Moravian-Silesian region there is not. This means that one type can include many small sites, but due to their average size, the total size is very small. This is clearly seen in the case of brownfields for housing, where the average size of a single site in Slovenia is 1.2 ha, while in the Moravian-Silesian region it is only 0.2 ha.

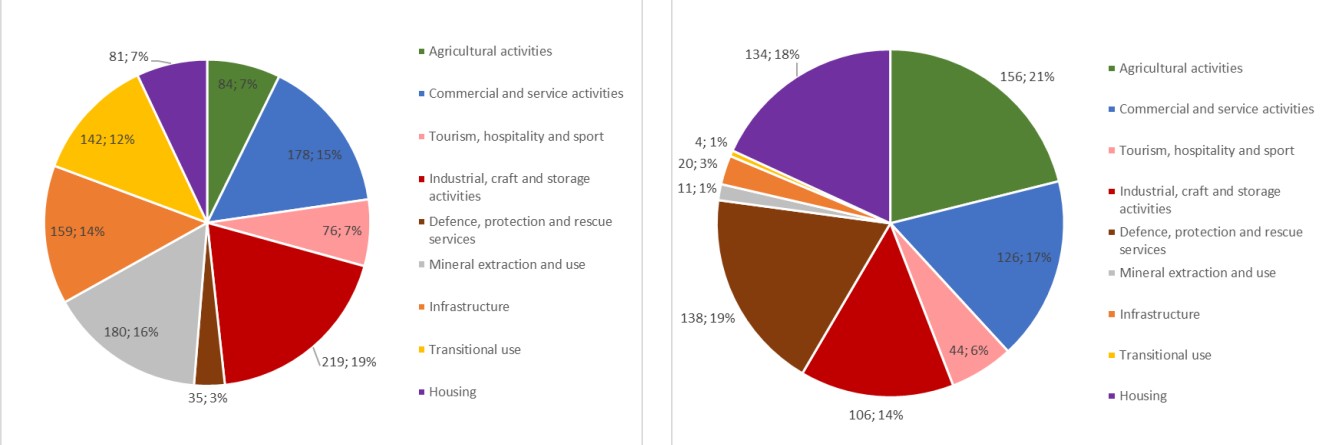

**Figure 3.** Brownfield structure in 2022 for Slovenia (**left**) and the Moravian-Silesian region (**right**) (by type). Based on [25,27]. Prepared by the authors.

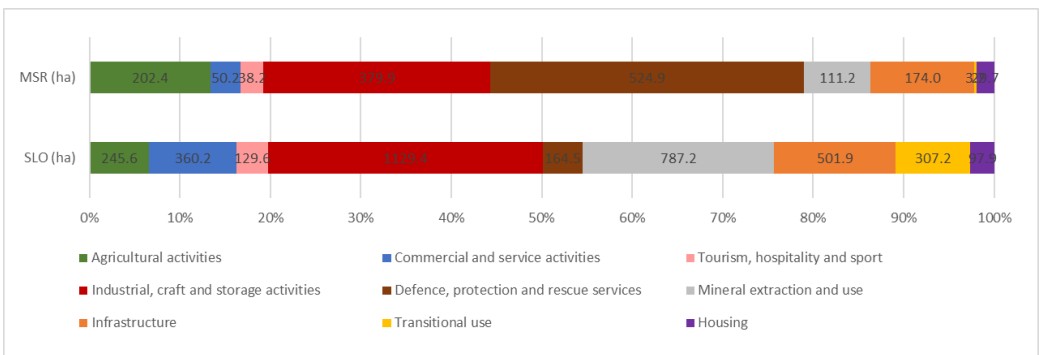

**Figure 4.** Regenerated brownfields by the type (2017–2022). Based on [24–27]. Prepared by the authors.

The structure of brownfield sites reflects the second reason. The Moravian-Silesian database shows that brownfields of industrial, craft and storage activities, as well as brownfields of defence, protection and rescue services represent the predominant type, including the classic large-scale military and industrial areas, which are the result of the original coal mining and steel industry in the region. The Slovenian brownfield structure reflects the different development of the past. In some parts of the country, we still find some traditional industrial and mining brownfields, but development has never created areas comparable in size to the Moravian-Silesian ones. The average size of Slovenian brownfields is 3.21 ha, which indicates that the country does not have really large, homogeneous sites and that large-scale investments in brownfields are often not possible or at least more difficult to realise.

### 3.3. Brownfield Regeneration Brings Functional Changes

The previously presented different data included in the brownfields databases in Slovenia and in the Moravian-Silesian Region also allow us to identify the transformation directions and functional changes of brownfields in the period 2017–2022.

The course of succession shows a certain regularity, as it was already found in the case of the Upper Silesian Industrial Region in southern Poland [14]. In the case of Slovenian and Moravian-Silesian brownfields, the new functions, which are close to the former land use, also predominate. Such solutions for brownfields are usually the simplest and cheapest and allow investors to make the best use of existing infrastructure [14].

Over the five-years period, a total of 187 brownfields were regenerated in Slovenia and 104 brownfields were regenerated in the Moravian-Silesian region (Figure 5). Although we found a big difference in the overall structure of brownfields in both case studies, the

structure of regenerated brownfields is much more similar. The structure of regenerated brownfields by type shows that the share of brownfields of industrial, craft and storage activities predominate (25% in Slovenia and 39% in the Moravian-Silesian Region), followed by brownfields of commercial and service activities (21% in Slovenia and 16% in the Moravian-Silesian Region) and brownfields for housing (17% in Slovenia and 16% in the Moravian-Silesian Region).

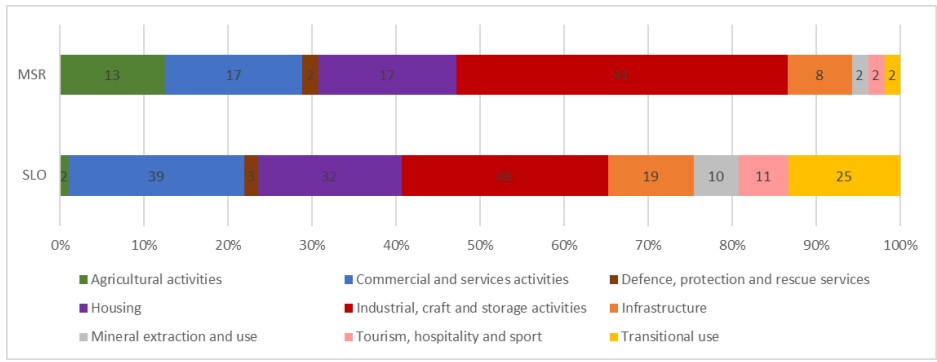

**Figure 5.** Comparison of brownfields in Slovenia (SLO) and the Moravian-Silesian region (MRS) by total area (ha) and by brownfield type. Based on [25,27]. Prepared by the authors.

The dominant share of regenerated industrial brownfields is to be expected due to their high share in the overall structure, but the significant share of regenerated commercial/service activities and housing sites shows the positive economic impulse and a vibrant construction sector present in several European countries. Several new investments, especially those of economic interest (e.g., the construction of new multi-family housing), are also being made on previously abandoned, underutilized land.

The strong economic impetus is evident in the flows of change shown in Figures 6 and 7. As noted earlier, several brownfield sites have been recently redeveloped for housing, but it is clear from the figures that housing function has been established on many other types of brownfield sites as well. Among the new functions in Slovenia, housing strongly dominates and is represented on 55 regenerated sites, followed by industry (41) and commercial and service activities (35).

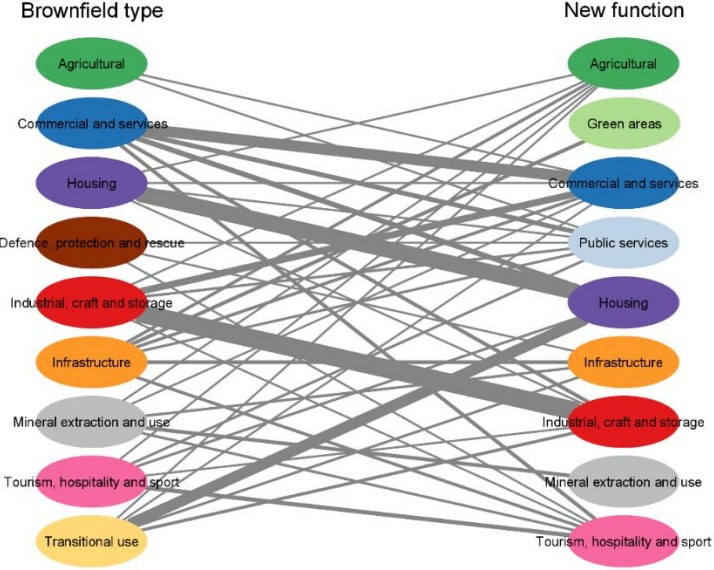

**Figure 6.** The currents of changes—brownfields regeneration in Slovenia (2017–2022). Line thickness indicating the number of changes. Based on [24,25]. Prepared by the authors.

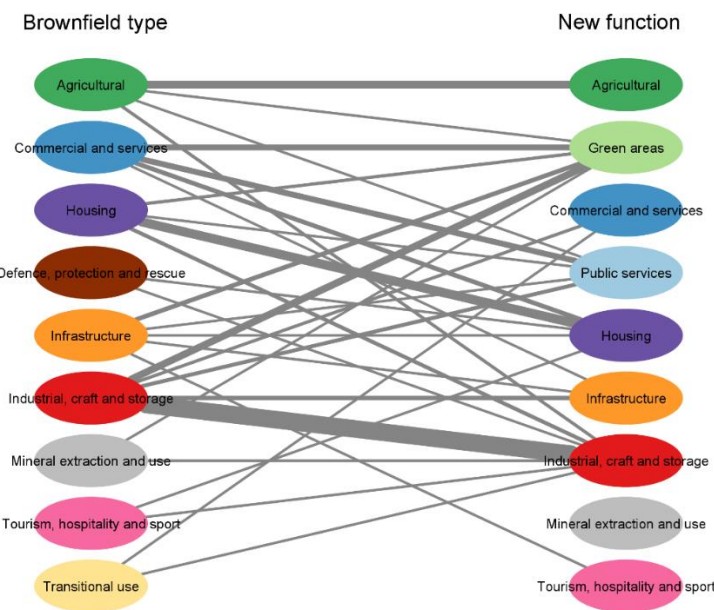

**Figure 7.** The currents of changes—brownfields regeneration in the Moravian-Silesian region (2017–2022). Line thickness indicating the number of changes. Based on [26,27]. Prepared by the authors.

As for the diversity of new functions on brownfields, the Moravian-Silesian region shows a slightly lower diversity (Figure 7), which we attribute to a smaller number of analysed regenerated brownfields and less detailed data collection. As far as new functions on regenerated brownfields are concerned, industry dominates (33), which can be directly associated with a lively economic pulse. Industry is followed by green areas (22) and housing (18).

In the period 2017–2022, the total area of regenerated brownfields in the Moravian-Silesian region was 421.5 ha and in Slovenia, 714.2 ha. Looking at the structure of regenerated sites in both case studies, brownfields of industrial, craft and storage activities predominate (346 ha in Slovenia, 270.6 ha in the Moravian-Silesian Region) (Figure 4). The average size of regenerated brownfields is in both cases very similar—3.82 ha in Slovenia and 4.05 ha in the Moravian-Silesian Region. We must emphasise that if we compare the average size of regenerated brownfields with the general average size of brownfields in 2022, we see that regenerated brownfields are significantly larger. This is even more characteristic for the Moravian-Silesian region.

Although the average size of regenerated brownfields is over 4 ha, Figure 8 shows that small brownfields predominate in the structure of regenerated sites (over 40%). This could simply be due to the fact that both Slovenia and the Moravian-Silesian region do not have large brownfields that are attractive to investors, or because large brownfields are often more complex to redevelop and require more time and higher financial investments.

Despite the fact that small brownfields predominate among the regenerated ones, the average size of a regenerated brownfield is larger in both case studies, which is a consequence of the fact that some very large regenerated sites are included. However, it is not enough to look only at the size, but also at the nature of these sites. In Slovenia, the large regenerated sites are dominated by abandoned or underutilised industrial zones whose construction was halted during the economic crisis and which subsequently became fully utilised due to economic growth. In the Moravian-Silesian region, on the other hand, these sites are large industrial areas whose creation is a consequence of the historical development of the region and whose regeneration is the result of the intensive restructuring of the region.

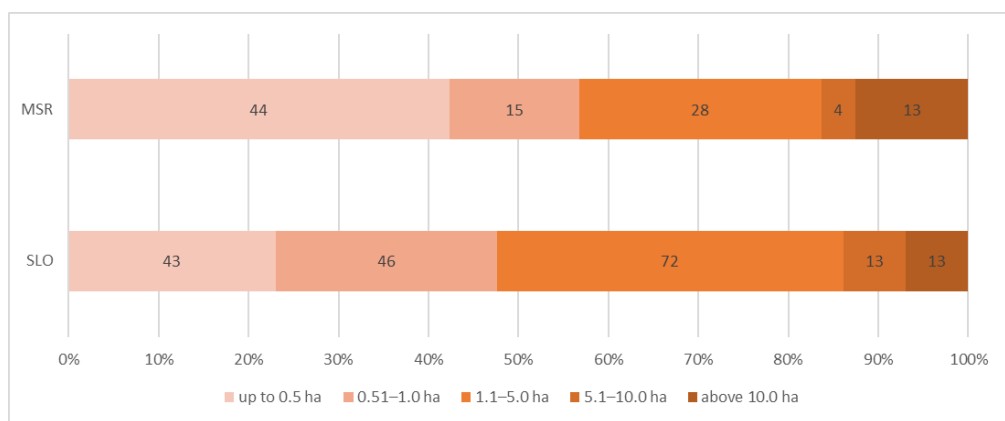

**Figure 8.** New forms of land use according to brownfield size. Based on [24–27]. Prepared by the authors.

Finally, the relationship between ownership and the regeneration process was also investigated. As it is known from the literature [55–58], the private investments are more efficient in the regeneration process. In both regions, regenerated brownfield sites that are privately owned account for more than half of all sites (Figure 9).

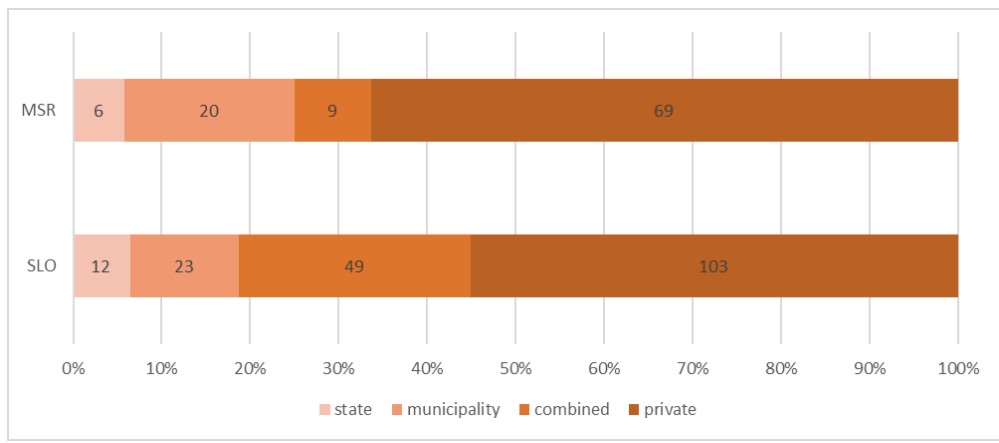

**Figure 9.** Ownership of regenerated brownfields. Based on [24–27]. Prepared by the authors.

Nevertheless, the proportion of regenerated brownfields in public ownership (state, municipality) is almost identical to that of private brownfields. Significant differences in ownership can be seen in the share of regenerated municipal brownfields (almost 20% in the Moravian-Silesian region) and a very high share of private brownfields (almost 70% in the Moravian-Silesian region). In Slovenia, the share of regenerated brownfields where ownership is combined (public-private) is significantly higher (over 25%). This is often the most difficult ownership structure, where the fewest sites are regenerated, and the causes are most often related to misaligned desires and interests.

Ownership has proven to be a very dynamic component that changes very quickly and can significantly affect the regeneration process, especially when it involves fragmented ownership, lack of owner interest, unequal interests, etc. Ownership often changes just before the regeneration process begins, making it an important factor in the onset of change—but the question remains as to what changes this will it bring.

Regarding the spatial distribution of regenerated brownfields, we also notice some similarities (Figures 10 and 11). There is a clear concentration of regenerated brownfield sites around Ljubljana and Ostrava, which is not a surprise, as these are regional centres, with generally higher concentration of brownfields. On the other hand, we can observe presence of regeneration also generally across both pilot areas.

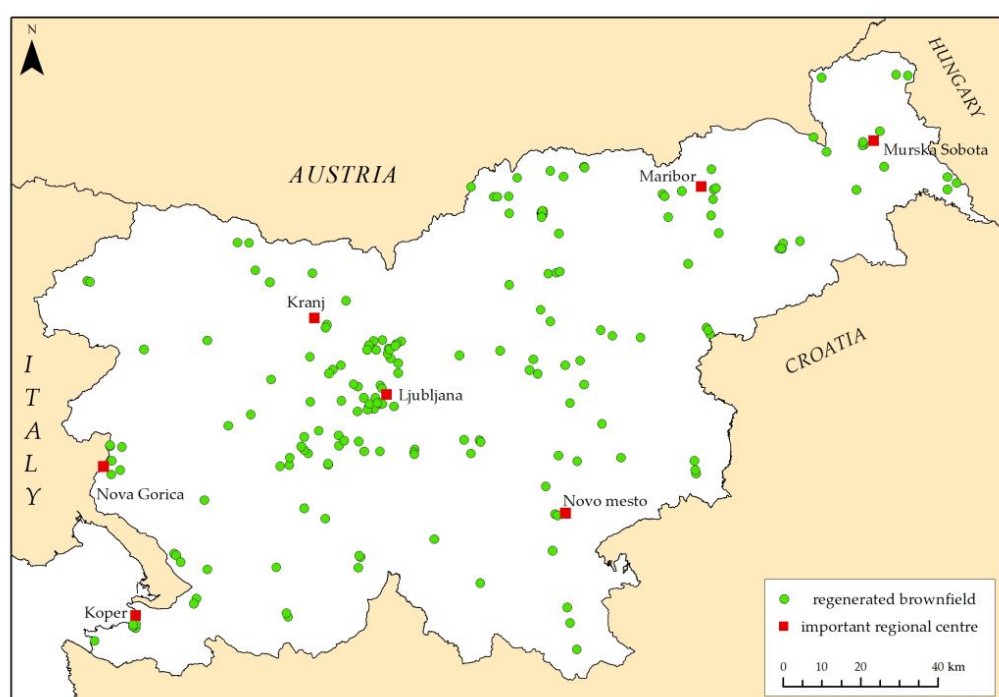

**Figure 10.** Spatial distribution of regenerated brownfields in Slovenia. Prepared by the authors [19].

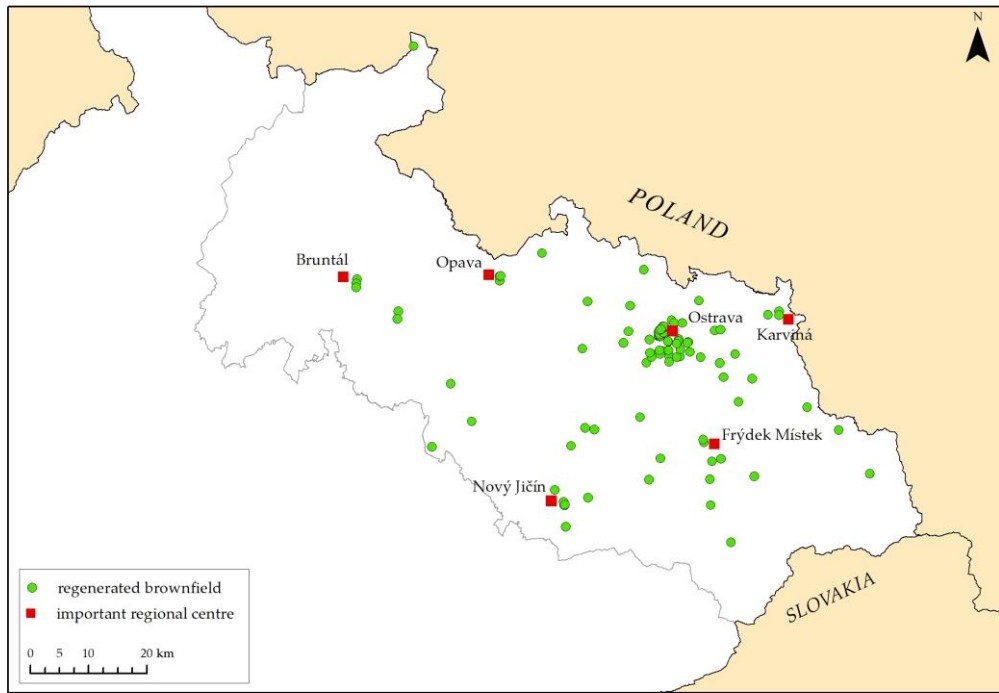

**Figure 11.** Spatial distribution of regenerated brownfields in the Moravian-Silesian region. Prepared by the authors [19].

## 4. Conclusions

The article presents an example of two countries with different historical development of brownfields databases, which, when analysed closely, reveals many related problems, but also points to the factors of success. In both cases, we note the importance of ensuring continuity of work and knowledge transfer in the management of brownfields and the detection of spatial changes. This has been ensured in Slovenia by the Faculty of Arts (University of Ljubljana) since 2015, and in the Moravian-Silesian region the responsible

organization is MSID. Over the years, data monitoring in the Moravian-Silesian region has been interrupted and changed, leading to errors and disruption of monitoring processes at sites.

Although the Slovenian database has only existed since 2017, it has made greater technological and methodological progress in a shorter period of time (compared to the Moravian-Silesian region). Its youth has proven to be an advantage, as the country has been able to draw on foreign experience and studies in developing its database, thus avoiding many potential errors. The result is a technologically advanced database. The comparison between Slovenia and the Moravian-Silesian region has also raised new questions concerning the criteria for inclusion of sites in the database. An example is the size of the brownfield. The minimum size criteria used by Slovenia often results in many small brownfield sites not being included in the database. Therefore, the question arises whether the criteria should be changed and adjusted in the future, as it has been done in the Czech Republic.

The article provides insight into a general picture of brownfield regeneration patterns in both case studies. In 2017–2022, there was a growing propensity to redevelop underutilised brownfield sites across Europe due to positive economic growth and a strong construction sector. Despite the fact that we are comparing different countries with different economic backgrounds, the focus of regeneration is on the same types of brownfields. This is clearly evident in the regeneration of brownfields for housing, as we can observe the intensive construction of residential neighborhoods on formerly abandoned construction sites and unfinished residential areas.

The structure of regenerated brownfields in both case studies is much more similar than the general structure of brownfields. Looking at the general structure, it is clear that it reflects the historical development of the two case studies, while a more detailed analysis of the regenerated sites shows that similar types of brownfields are the focus of regeneration.

Despite some peculiarities, the flows of change in brownfields in the two case studies show a high degree of dispersion of new activities occurring in regenerated sites, which is slightly higher in the Slovenian example. This indicates the lack of systematic planning for brownfield regeneration and the absence of strategies. Regeneration is therefore often planned in the short term and left to random investments by private investors, a fact that is further emphasised by the ownership structure of the regenerated sites. It is important to emphasise that the Moravian-Silesian region has an advantage here. Having a longer tradition in dealing with brownfields, it has already managed to set up the *Action plan for brownfield regeneration* based on a project approach, while Slovenia lags behind.

Ideally, brownfields should be regenerated as part of the same activity [14], but the dispersion of flows clearly shows that this is not the case. The lack of guidelines and rules in Slovenia leads to a more economically oriented recovery, driven by market demand [59]. It is clear that each country should have a responsible body for managing the brownfields database and updating the data, as this is the only way to ensure consistency of information, transfer of knowledge and successful management of the sites, which is ultimately the key to their regeneration.

In further research it will be necessary to focus not only on the database and its structure, but also on the legislative background of the whole regeneration process (environment, building legislation, etc.). It will be necessary to focus on the spatial and urban planning processes and to look closer at the degree of responsibility of individual stakeholders in the regeneration process.

Countries (and spatial planners) should approach redevelopment differently and see long-term site regeneration as key to more sustainable redevelopment and spatial planning. New activities on sites should be better thought-out and take into account the real needs of local communities, as this will ensure better acceptance of the new activity in the local environment and its long-term existence [59]. This is a key step for the successful management of brownfields, and in particular can contribute to a successful land recycling process.

**Author Contributions:** Conceptualization, B.L., B.V. and L.R.; methodology, B.L., B.V. and L.R.; software, L.R.; validation, B.L., B.V. and L.R.; formal analysis, B.L., B.V. and L.R.; investigation, B.L., B.V. and L.R.; resources, B.L., B.V. and L.R.; data curation, B.L., B.V. and L.R.; writing—original draft preparation, B.L., B.V. and L.R.; writing—review and editing, B.L., B.V. and L.R.; visualization, L.R.; supervision, B.L. and B.V; project administration L.R.; funding acquisition, B.L. All authors have read and agreed to the published version of the manuscript.

**Funding:** The research was supported by the Slovenian Research Agency (P6-0229 Sustainable regional development of Slovenia).

**Data Availability Statement:** The data presented in this study are available on request from the corresponding author. The data are not publicly available, as they are only available in Czech and Slovenian language, and contain personal comments.

**Acknowledgments:** The authors would like to thank MSID for technical support and access to the Moravian-Silesian database, and the Ministry of Economic Development and Technology (Republic of Slovenia) for financing the Slovenian database update in 2019.

**Conflicts of Interest:** The authors declare no conflict of interest.

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
