# Peer review of "Brownfield Data and Database Management—The Key to Address Land Recycling"

_land, doi:10.3390/land12010252_

Round 1

Reviewer 1 Report

This work is an interesting comparison between two existing brownfield databases, one in Slovenia and one in the Czech Republic. In my review, I found the authors provide a detailed comparison of a newer database created in Slovenia, and the ways in which it is more advanced than the older database created for a segment in the Czech republic. The paper could be improved by focusing on how the modern Slovenian database improves outcomes. While the authors state the reasons why the Slovenian database is "better", the paper does not inform the reader whether this improvement leads to quantifiable better/faster remediation than the comparison database. The authors provide an analysis of the brownfields from the database, but do not convince the reader that the Czech republic is suffering from reduced brownfield remediation efficiency due to their dated design (though it is my belief that it likely does).

The paper is well researched and thorough, but could benefit from defining some terms more clearly, such as: "efficient use of land", "rational and efficient development", and most importantly what it means for the flows of brownfield remediation to have a "high degree of dispersion".

I think the weakest portion of the paper is the last two paragraphs, where the authors provide many normative policy prescriptions on how places should better their remediation programs by "long-term site regeneration", and that private remediation practices lead to problems such as "mixing of activities that are not compatible with each other". If market outcomes leads to an diverse mix of business, residences, and industry, how do we judge this as inefficient? These policy prescriptions really would benefit from either some source/paper which would tell me why, but I am not convinced the comparison of the two databases by the authors leads me to agree with these statements.

Overall, I think this is a solid work that just needs some greater context for the comparison of the databases, some tightening of the written work (definitions, clarity of writing), and either a greater connection between the main body of the work with the policy prescriptions, or removal of these normative statements. 

Author Response

Thank you very much for your stimulating comments. Also, thank you very much for the time you spent on the review. The responses are detailed in the attached document. We hope that we have been able to process your comments so that the article is acceptable for publication. 

Reviewer 2 Report

Abstract

The author should refer to the most significant results of the application of his or her research method in such a way as to arouse the reader's interest.

Introduction

The author rightly points out the problem of the diversity of the two contaminated sites considered in the research, but does not provide any description of the sites, nor does he indicate what methods were used. I would like to advise the author, to say explicitly, of the different disused industrial sites, to describe the characteristics and differences of each of them, and to indicate, referring to the rehabilitations taken into consideration, what are the positive and negative points of each of them, and then to indicate the path that he has taken in this study and then to justify his method of rehabilitation of industrial sites. The introduction should be more detailed in trying to present the rehabilitation references in the literature. Therefore, a broader review of the literature more focused on its area of study would be appropriate.

Material and method

In introducing the two sites, the author states that the data are not homogeneous because Slovenia has only recently rehabilitated industrial sites, while the Czech Republic has a larger archive because it has been rehabilitating for more years, which implies a difference in addition to the geographical differences. The author should explain why he conducted this research in these two seemingly very different territories and why this might invalidate the data.

Regarding Table 1 of the methodology, it is unclear what is indicated by brownfield types/brownfields if only brownfield types are then listed. It would be more appropriate to leave only brownfield types in the list.

In Table 2, in the indication of the second column, the description should be left and the examples deleted as it is confusing.

The author is advised to present the steps of the methodology with a figure indicating for each step what has been done and what has been taken into account.

In addition to the table for Slovenia and the Czech Republic, it is advisable to present a map of the sites studied, so that the reader knows where these sites are located in the country. For example, it would be useful to know whether these contaminated sites are located near urban centers or in rural or expanding areas. For each of these sites, depending on their geographic location, the degree and objectives of the remediation conducted may depend. In other words, an industrial site near an urban center will be remediated differently than a site located in a geographic area far from an urban center. Similarly, it can be said that the location of each industrial site also modifies the objectives of the remediation

It is recommended that the author revise the paragraph with these considerations and present a map of the sites for Slovenia and the Czech Republic and include a table of the rehabilitation goals achieved. This would be very interesting to understand. It would then be a question of whether the interventions are of a public or private nature.

Figure 2: no presentation In the text for Figure 2, it is not clear where the percentages in the graph come from. It would be appropriate for the author to explain the percentages in the graph when presenting it. The reference is in section 3.2 . The figure should be placed immediately after the reference text.

Table 4 too much text in the boxes; find a way to summarize the data with reference to the document under consideration. A table should contain a summary or easily readable data. The author must find the most effective way to present the table with less text. In addition, the table should include a description of the column heading on the second page.

3.2 results of the comparison

In the text, reference is made to figure 2 but in reality the data are contained in figure 3. It is advisable to correct the error because the reader has to go back and look at figure 2 . Very complicated to understand. Put the text and figures in order.

It is difficult to understand and relate the text to the figures that are not presented first by the author. Please present the figures and then insert the reference figure. Very confusing, please ask the author to rewrite the entire results and figures paragraph with this point in mind.

Conclusions

The conclusion, while criticizing the study of the two sites and pointing out their differences, should indicate in another paragraph the weaknesses of the research and how to improve this handicap. It should also leave room for future action, or at least say or give an opening for further work and research to be done. The author should tell us his opinion, what other researchers should do to improve the rehabilitation of contaminated sites.

Author Response

Thank you very much for your stimulating comments. Also, thank you very much for the time you spent on the review. The responses are detailed in the attached document. 
We have attempted to better explain the objectives of our research. We have also attempted to better explain the concept of brownfields.
We hope that we have been able to process your comments so that the article is acceptable for publication. 

Round 2

Reviewer 2 Report

Authors are advised not to use personal forms for example "We have, etc." but impersonal forms ...